The significance of information variables in polydrug use by adolescents: insights from a cross-sectional study in Tarragona (Spain)

de Andrés-Sánchez Jorge jorge.deandres@urv.cat
Belzunegui-Eraso Angel
Valls-Fonayet Francesc
University Rovira i Virgili , Tarragona , Spain
Musetti Alessandro
Electronic publication date: 2024 Jan 19
Publication date: 2024
Volume: 12
Electronic Location ID: e16801
Received 2023 Nov 1; Accepted 2023 Dec 26
Copyright: © 2024 de Andrés-Sánchez et al.
Copyright year: 2024
Copyright holder: de Andrés-Sánchez et al.
License: This is an open access article distributed under the terms of the Creative Commons Attribution License, which permits unrestricted use, distribution, reproduction and adaptation in any medium and for any purpose provided that it is properly attributed. For attribution, the original author(s), title, publication source (PeerJ) and either DOI or URL of the article must be cited.
License URL: https://creativecommons.org/licenses/by/4.0/

Keywords: Substance use, Polydrug use, Health literacy, Adolescence, Monitored information sources, Non-monitored information sources

Funding: Ministerio Español de Ciencia e Innovación, Plan nacional R+D+I 2019 PID2019-104310RB-C21 This article is a result of the research project: “Prevention of drug use and gambling in adolescents: the information paradox. The case of Tarragona”, which has been funded by Ministerio Español de Ciencia e Innovación, Plan nacional R+D+I 2019. Code: PID2019-104310RB-C21. The funders had no role in study design, data collection and analysis, decision to publish, or preparation of the manuscript.

==============================
Substance use, especially among adolescents, is a significant public health concern, with profound implications for physical and psychological development. This study aimed to evaluate the quantity and sources of information available to adolescents regarding polydrug use. A cross-sectional survey was conducted in Tarragona involving adolescents with an average age of 16.44 years. This study assessed the number of substances used (alcohol, cigarettes, and cannabis) in the past month, along with information sources related to substance use. Monitored sources (e.g., schools, parents, and mass media) and unmonitored sources (e.g., peers, siblings, internet) were distinguished. In addition, four individual and four environmental control variables were considered. Multinomial logistic regression analysis revealed that incorporating variables related to adolescents’ substance use information and its sources enhanced the explanatory model, surpassing control variables. The degree of information about substance use did not significantly explain consumption patterns, but the number of information sources, both monitored and unmonitored, did. The unmonitored sources were associated with increased polydrug use. Conversely, greater reliance on supervised sources for information was linked to reduced single-substance and polydrug use. This protective effect increased with an increase in the number of substances used. In conclusion, information obtained from monitored sources acts as a deterrent to substance consumption, consistent with findings suggesting that greater health literacy among adolescents discourages substance use. Conversely, this study suggests that information from more informal sources may encourage heavier polydrug use, aligning with reports indicating that adolescents with a more comprehensive understanding of substance use consequences tend to engage in heavier drug use.

Introduction

Adolescence is a pivotal phase in human development, and substance consumption is a prominent concern among teenagers (Leal-López et al., 2021). This period involves engaging in risky behaviors and making impulsive choices, often exposing adolescents to psychoactive substances. In this early stage of life, substance use can influence brain development and potentially increase the risk of long-term addiction (Whyte et al., 2018). The three most widely used substances among adolescents are alcohol, tobacco, and cannabis (Gray & Squeglia, 2018; OEDA (Spanish Observatory on Drugs and Addictions), 2022).

These substances have neurotoxic effects on both cognitive and mental functions. These disorders contribute to the development of personality disturbances (Magallon-Neri et al., 2015), mental disorders, depression, and attentional deficits. There is some evidence, especially in the case of cannabis, that its use might increase the likelihood of its future use due to a gateway effect (Hamidullah et al., 2020; Shapira et al., 2020). Furthermore, substance use often results in decreased academic performance, risky behaviors, accidents, and undesirable behaviors, such as sexual harassment (Francis et al., 2019; Livingston et al., 2022). When discussing polysubstance use, we refer to the consumption of two or more drugs, whether legal or illegal, within the same period of time (Brière et al., 2011). Alcohol is the most frequently used substance in simultaneous polysubstance use patterns and is often accompanied by tobacco and cannabis (Brière et al., 2011; Jongenelis et al., 2019; OEDA (Spanish Observatory on Drugs and Addictions), 2022). Mixing substances are considered to be a particular risk to the safety and well-being of young people, especially because of their potentially unpredictable additive or interactive effects (Brière et al., 2011).

Multiple variables often influence substance use through complex interactions. These factors can encompass genetic predisposition, personality traits, parental style, levels of school engagement, peer influences, cultural factors, etc. (Cross, Lotfipour & Leslie, 2017; Trucco, 2020; Nawi et al., 2021). In this context, although several studies have noted the importance of health literacy in promoting healthier lifestyles (Jayasinghe et al., 2016; Bröder et al., 2017), one of the less explored factors is the influence of adolescents’ information regarding substance use (Belzunegui-Eraso et al., 2020).

Several authors have reported that health literacy protects against substance use in young people and teenagers (Fleary, Joseph & Pappagianopoulos, 2018; Sadeghi et al., 2019; Rolova, Gavurova & Petruzelka, 2021). However, neither the information campaigns carried out in educational centres nor those carried out in the mass media necessarily have a significant effect (Mélard et al., 2020). Dermota et al. (2013) in Switzerland and Belzunegui-Eraso et al. (2020) in Spain reported that more information about substance use among adolescents was associated with increased tobacco and cannabis consumption. Belzunegui-Eraso et al. (2020) call this the “information paradox” of substance use. One reasonable argument supporting this finding is that health literacy depends on the trustworthiness of information sources (Chen et al., 2018; Buawangpong et al., 2022). For this reason, Belzunegui-Eraso et al. (2020) suggested further analysis of the influence of information sources on adolescent substance use.

In this study, we considered six sources of information that are commonly regarded as influential on health literacy, specifically among young people: schools, parents, media, peers, siblings, and the Internet (Manganello, 2008; Dermota et al., 2013; Bujnowska-Fedak, 2015; Chen et al., 2018; Buawangpong et al., 2022). The reliability of each item can be differentiated by the level of regulation provided by public bodies and professional associations regarding the information they provide to young people.

In Spain, the information provided by schools, parents, and the media is subject to strong regulations and monitoring. This information usually discourages substance use and emphasizes the harm associated with consumption. While there is limited assessment of the effectiveness of many drug supply control policies and, as a result, their actual impact is often not measured (Strang et al., 2012), there is some evidence suggesting that the perception of the potential harm of drug use may deter certain adolescents from using them (Zimmerman & Farrell, 2017).

First, educational institutions, particularly schools, are paramount for imparting knowledge to young people. They serve as the foundational pillar of health literacy, which is a crucial aspect of youth education (Manganello, 2008). The primary goal of education, as emphasized by UNICEF (2021), is to foster the health and well-being of youth, with health literacy being a key contributor (Fleary, Joseph & Pappagianopoulos, 2018; Sadeghi et al., 2019). In Tarragona, public institutions and healthcare professionals closely monitor substance abuse prevention programmes in schools.

Second, parents bear a significant responsibility for their children’s welfare, as mandated by international and national laws (United Nations, 1989; Spanish Civil Code, 1889). Fulfilling this obligation includes providing reliable information regarding the adverse effects of substance use. Spanish law even sanctions against the loss of custody when parental behavior endangers a child’s well-being due to issues such as substance abuse (Odériz-Echevarria, 2023).

Finally, conventional media channels, which are tightly regulated by legislation and ethical codes, serve as a means of disseminating information about drug risks. Stringent bans on tobacco and alcohol advertising and sponsorship exist at both the European and national levels (European Union Directives, 1989, 2003; Feliu et al., 2019). Similarly, marijuana and hashish are illegal drugs; therefore, no advertisements are allowed. While these outlets are used by public health authorities to warn about tobacco dangers, their content is subject to scrutiny for its accuracy (Dermota et al., 2013). Furthermore, while the ethical code of the Spanish College of Journalists safeguards, among other issues, the accuracy of information provided in the media (Federation of Associations of Journalists of Spain, 2017) and the content offered by television during protected hours for children are subject to special oversight in Spain by the National Commission of Markets and Competition (2015).

Certain sources of information have limited control over their influence on adolescents, potentially encouraging substance consumption. Peer-, sibling-, and Internet-based information channels lack effective oversight from both legislative and administrative perspectives.

Peers and siblings often embrace a perspective that prioritizes immediate gratification and hedonism over long-term risks and may focus on themes such as enhancing social enjoyment, relaxing, or associating substance use with status and glamour (Megías et al., 2006; Eisenberg et al., 2014; Montgomery et al., 2020; Gupte, D’Costa & Chaudhuri, 2020; Henneberger, Mushonga & Preston, 2021). This can lead to adolescents underestimating the potential harm associated with substance use. To understand the trends in adolescents’ risk-taking attitudes, it can be helpful to consider the justification of the so-called dual-system model. According to this framework, risk-taking is attributed to the activation of an early maturing socioemotional incentive processing system, which intensifies adolescents’ attraction to exciting, pleasurable, and novel activities. This occurs concurrently with an immature cognitive control system that has not yet been sufficiently developed to consistently inhibit potentially hazardous impulses (Shulman et al., 2016).

The Internet has the potential to provide valuable health-related information and support networks. It allows access to substance-use prevention information (Parissi-Poumian et al., 2023) and facilitates online interventions involving professionals, peers, or a combination of both (Ahmad et al., 2022). However, this information often lacks control (Tonsaker, Bartlett & Trpkov, 2014), which can have a negative impact on health literacy (Chen et al., 2018). Furthermore, alcohol use serves as a source of health misinformation, with one of the primary areas of concern being drug and tobacco consumption (Suarez-Lledo & Alvarez-Galvez, 2021), which may also stimulate risky behaviors (Vannucci et al., 2020). Adolescents may encounter content that glamorizes substance use by peers or influencers (Gateway Foundation, 2023), stumble upon scientifically unfounded information, or even gain access to legal substances without proper regulation (Gateway Foundation, 2023).

The reflections presented in the previous paragraphs motivate the current study, which should be contextualized in the most common polysubstance use, namely, the combination of alcohol, tobacco, and cannabis. Specifically, we address the following research question (RQ): What are the protective and facilitating factors of polysubstance use, and what influence do information factors have? In this regard, we tested the relevance of the level of information that adolescents perceived and the number of sources of information consulted monitored versus unregulated on the prevalence of polysubstance use. Therefore, whereas we expect that monitored sources have a protective effect on polysubstance use, nonmonitored sources can stimulate this practice.

Materials and Methods

Materials

This study was conducted using a cross-sectional survey conducted between March and April 2021. The survey collected data from a structured questionnaire administered to secondary school students in Tarragona, Spain. The participants were teenagers in their last year of compulsory primary education, either in the two possible years of compulsory secondary education or enrolled in vocational training. To conduct this study, we obtained permission and assistance from school principals with the help of social workers from the Tarragona City Council. The survey included 66 questions and was completed online, taking approximately 15–25 min for each participant.

This study utilized a sample of 1,307 observations. The sample was obtained from a broader population of approximately 8,000 teenagers, ensuring a margin of error below 3% (Conroy, 2016). Table 1 illustrates that a significant proportion of the individuals, specifically 700 (53.56%), were either 16 years old or younger. Similarly, 573 responses (43.84%) were obtained from adolescents aged 17 years or older. The average age of the adolescents was 16.44 years, with a standard deviation of 0.9 years. Regarding sex distribution, 608 responses (46.52%) were from females, and 669 responses (51.19%) were from males.

Table 1 Characteristics of the analysed sample of adolescents (N = 1,307).

Gender	Age	
Females: 608 responses	>= 17 years: 573 responses	
Males: 669 responses	<= 16 years 700 responses	
NA: 30 responses	NA: 34 responses	
	Mean = 16.44. Standard deviation = 0.96 years.	
The adolescent lives with
at least 1 parent 1,186
without parents: 75
NA: 3	Place where was born:
The adolescent
Spain: 1,150 (87.99%); Abroad: 152 (11.63%); NA: 5 (0.38%)
Both parents were born:
Spain: 879 (67.25%); Abroad 299 (22.88%): Only one parent in Spain 129 (9.87%)	

Among the adolescents, a vast majority (approximately 95%) resided with at least one parent and were born in Spain (1,150, 87.99%). Similarly, 879 respondents (67.25%) reported that both parents were born in Spain, whereas 299 indicated that both parents were born abroad. In addition, 129 respondents (9.87%) reported having one parent born abroad.

With respect to the institutional review board statement, (1) the applicable regulations were respected, which consisted of Regulation (EU) No. 2016/679 of the European Parliament and of the Council of April 27, 2016, on the protection of natural persons with regard to the processing of personal data and on the free movement of such data and Organic Law 3/2018 of December 5 on the Protection of Personal Data and Guarantee of Digital Rights; (2) all participants and their legal guardians were provided with detailed written information about the study and procedures; (3) the anonymity of the collected data was ensured at all times; (4) approval was obtained from the board or ethics committee of the researchers’ institution (CEIPSA-2021-PRD-0039); and (5) completion of the questionnaire was conducted only after obtaining consent for the use of the data in the research. Verbal informed consent was obtained from all subjects involved in the study.

Measurement variables

To assess the research question, we performed a regression analysis to relate the number of substances consumed in the last 30 days to eight control variables and three linked variables with the information that the adolescents received about substance use. The outcome variable was defined as the number of substances consumed over 30 days (NUMSUBST) divided by the sum of the substances consumed over the last 30 days, as displayed in Table 2 (alcohol, cigarettes and cannabis). Therefore, NUMSUBST takes the values {0, 1, 2, 3}. It is important to note that when stating the values of NUMSUBST, we considered the number of different substances used within the last 30 days, which is not necessarily the maximum number of substances consumed simultaneously in one moment during the measurement period.

Table 2 Questions of the survey used to develop our study and descriptive statistics.

Input questions		
Q1: What is your sex?	1.- Girl, 2.- Boy, 3.- I prefer not answer	
Q2: What is your age?							
Q3: Non adherence to norms	(1)	(2)	(3)	(4)	(5)	NA	
NORMS.1: Most rules can be broken if they are not convenient.	182	256	456	172	114	127	
NORMS.2: I follow the rules that I want to follow.	145	193	294	364	212	99	
NORMS.3: It is hard to trust anything because everything changes.	63	96	385	384	256	123	
NORMS.4: In fact, no one knows what is expected of him/her in life.	57	94	313	405	325	113	
NORMS.5: You can never be sure of anything in life.	73	87	243	413	397	94	
NORMS.6: Sometimes, it is necessary to break the rules to succeed.	109	130	391	308	266	103	
NORMS.7: Following the rules does not guarantee success.	63	87	341	331	383	102	
Q4: AGRESSIVE	(1)	(2)	(3)	(4)	(5)	NA	
AGRESSIVE.1: I have been easily bothered or irritated	170	269	350	284	175	59	
AGRESSIVE.2: I have had outbursts of anger that I could not control.	432	349	215	154	96	61	
AGRESSIVE.3: I have wanted to break or damage things.	551	316	180	108	90	62	
AGRESSIVE.4: I have had a fight with someone.	670	279	173	54	66	65	
AGRESSIVE.5: I yelled at someone or threw things at them	649	279	172	74	66	67	
Q5: School support	(1)	(2)	(3)	(4)	(5)	NA	
SCHOOLENG.1: The adults at my school care about me.	227	467	393	116	59	45	
SCHOOLENG.2: I have friends at my educational center who care about me.	616	441	141	40	28	41	
SCHOOLENG.3: The students at my educational center are kind to each other.	230	429	405	127	76	40	
SCHOOLENG.4: My educational center is helping me achieve goals that matter to me.	200	420	402	152	80	53	
SCHOOLENG.5: I enjoy participating in activities at my educational center.	202	367	452	139	97	50	
Q6: Parental support	(1)	(2)	(3)	(4)	NA		
PARSUPP.1: I receive care and affection from my parents.	37	62	248	893	67		
PARSUPP.2: With my parents, I can talk about personal matters.	129	246	374	475	83		
PARSUPP.3: I receive advice from my parents regarding my studies.	60	106	321	752	68		
PARSUPP.4: I receive advice from my parents regarding other topics (your projects).	70	131	340	694	72		
PARSUPP.5: I also receive help from my parents with other things.	49	99	302	781	76		
Q7: Parental control	(1)	(2)	(3)	(4)	NA		
PARCONT.1: My parents consider it important that my studies go well.	8	36	347	856	60		
PARCONT.2: They establish clear rules about what I can do at home.	55	114	547	502	89		
PARCONT.3: They establish clear rules about what I can do outside the house.	65	136	520	492	94		
PARCONT.4: They establish clear rules about when I have to be home in the evening.	95	184	473	448	107		
PARCONT.5: They know who I am with at night.	46	81	302	772	106		
PARCONT.6: They know where I am at night.	38	57	287	825	100		
PARCONT.7: They know my friends.	43	90	344	755	75		
PARCONT.8: They know the parents of my friends.	139	226	457	379	106		
Q8: Peer influence	(1)	(2)	(3)	(4)	(5)	NA	
PEERINFL.1: Sometimes you have to smoke cigarettes to avoid being left out of the peer group.	1,086	83	46	15	23	54	
PEERINFL.2: Sometimes it is necessary to drink alcohol to avoid being left out of the peer group.	998	129	80	25	17	58	
PEERINFL.3: Sometimes it is necessary to smoke cannabis to avoid being left out of the peer group.	1,111	57	43	18	20	58	
PEERINFL.4: Sometimes it is necessary to skip classes to avoid being left out of the peer group.	1,053	108	54	7	24	61	
Q9: INF_LEVEL	(1)	(2)	(3)	(4)	(5)	NA	
	29	42	137	480	550	69	
My information about
substance use come from:	Yes	No	NA				
Q10: School	866	350	91				
Q11: Parents/legal guardians	815	392	100				
Q12: Mass media	720	484	103				
Q13: Internet	861	356	90				
Q14: Siblings	306	892	109				
Q15: Peers and friends	584	616	107				
Number of sources	0	1	2	3	NA		
Monitored sources (Q10 + Q11 + Q12)	121
(9.26%)	251
(19.20%)	393
(30.07%)	451
(34.51%)	91
(6.96%)		
No monitored sources (Q13 + Q14 + Q15)	210
(16.07%)	449
(34.35%)	375
(28.69%)	183
(14.00%)	90
(6.89%)		
Prevalence last 30 days	No	Yes	NA				
Use 1: Did you use alcohol drinks last 30 days?	763
(58.38%)	502
(38.41%)	42
(3.21%)				
Use 2: Did you use cigarettes last 30 days?	1,071
(81.94%)	196
(15.00%)	40
(3.06%)				
Use 3: Did you use cannabis last 30 days?	1,078
(89.24%)	88
(7.28%)	42
(3.48%)				
Output	0	1	2	3	NA		
NUMSUBS = Number of substances (alcohol, tobacco and cannabis) consumed last 30 days	681
(52.10%)	380
(29.07%)	117
(8.95%)	57
(4.36%)	72
(5.51%)		
Notes:

(A) Quantities come in absolute value. (B) In the case of Q3, Q5 and Q8 (1) Completely disagree (2) Mostly disagree (3) Neither agree nor disagree (4) Mostly agree and (5) Completely agree. (C) In the case of Q4, (1) almost never; (2) rarely; (3) sometimes; (4) often; (5) almost always. (D) In the case of Q6: (1) very difficult, (2) difficult, (3) easy; (4) very easy. (E) in the case of Q7, (1) Does not apply at all to me; (2) Does not apply well to me; (3) Applies quite well to me; (4) Applies very well to me.

Among the control variables, four were individual factors, and four were environmental factors. The variables of interest, which are linked with adolescent information, quantify the volume of information available to adolescents and the number of supervised and nonsupervised sources that are at the origin of such information.

Variables related to the intrinsic factors of adolescents constitute the first group typically considered in the analysis of adolescent drug consumption (Trucco, 2020). In addition to more obvious factors, such as age and sex, aspects related to adolescents’ temperament (Trucco, 2020) play a crucial role in adolescents’ inclination toward substance use. In particular, adolescents exhibit emotional states (Carceller-Maicas et al., 2020) and certain attitudes, such as questioning norms (Nawi et al., 2021). Aggressiveness is of particular interest because of its strong association with substance use, as individuals with higher levels of irritability and impulsivity may turn to substances to alleviate these mood states (Kivimäki et al., 2014; Picciotto et al., 2015; Turner et al., 2018; Francis et al., 2019; Nawi et al., 2021). Likewise, it is well known that teenagers’ general tendency to question rules, including those related to drug consumption, can encourage such use (Boddington & McDermott, 2013; Kwon et al., 2018; Nawi et al., 2021). Hence, we used four control variables associated with the individual characteristics of the adolescents, defined based on the questions in Q1, Q2, Q3, and Q4 in Table 2, as shown in Table 3.

Table 3 Measurement of input variables.

Individual variables	
FEMALE: Takes a value of 1 if the response comes from a female and 0 otherwise.
AGE: Its value is 1 if it comes from a 17-year-old or older person and 0 otherwise.
NORMS: This variable comes from indicators in Q3 of Table 2 and measures the tendency of the adolescent to break the rules. It is built up as the standardized factor score of the indicators of the scale,
AGRESSIVE: This variable is built up with the indicators of Q4 of Table 2 and measures the irritability of teenagers. It is the standardized factor score of the indicators of the scal).	
Environment variables	
SCHOOLENG: Measures the adolescents’ emotional link and engagement with the center where they study. It is the standardized factor score of the 5 questions of Q5 (see Table 2).
PARSUPP: It measures parental support perceived by the adolescent. It is the standardized factor score of the indicators of scale Q6 in Table 2.
PARCONT: It fits the control over the teenagers’ activities exercised by parents. It is built up from the standardized factor score of the items in Q7 in Table 2.
PEERINFL: Measures the influence that teenagers’ peers have on them. It is built up from the indicators in Q8 and is fitted as the standardized factor score of the indicators of the scale (see Table 2).	
Information variables	
INF_LEVEL: Evaluates the extent to which teenagers feel that are informed about the repercussions of substance use. It is the normalized value of IQ9, calculated as INF_LEVEL = (IQ9-1)/4.
SUP_S: The normalized value of the number of supervised information sources. From Table 2 we state: (Q10 + Q11 + Q12)/3.
NON_SUP_S: The normalized value of the number of nonsupervised information sources. From Table 2 we state: (Q13 + Q14 + Q15)/3.	

With respect to the control variables related to adolescents’ environment, we considered school support, parental control and support, and peer influence, which are the main factors affecting adolescents’ microsystems (Trucco, 2020). School engagement is commonly accepted, and establishing emotional links with educational institutions serves as a protective factor (Rovis, Jonkman & Basic, 2015; Tomczyk, Isensee & Hanewinkel, 2015; Vogel et al., 2015; Jongenelis et al., 2019).

It is well documented that permissive attitudes of parents toward substance use increase the risk of substance use during adolescence, while parental disapproval acts as a protective factor (Kristjansson et al., 2008; Ozer et al., 2013; Kim-Spoon et al., 2014; Berge et al., 2016; De Looze et al., 2017; Sharmin et al., 2017; Stanley, Swaim & Dieterich, 2017; Staff & Maggs, 2020; Trucco, 2020; Cavazos-Rehg et al., 2021; Mehanović et al., 2022). Although parental control discourages substance use, authoritarian parenting can widely be recognized to facilitate substance use (Trucco, 2020). Conversely, affectionate and democratic parents who allow adolescents to feel supported and respected provide a shield against drug use (Kristjansson et al., 2008; Ozer et al., 2013; Trucco, 2020).

A third pillar in teenagers’ mycrosystem is the influence of peers on substance consumption (Trucco, 2020; Henneberger, Mushonga & Preston, 2021). Many studies have shown strong links between alcohol, tobacco, cannabis, and peer influence (Cengelli et al., 2012; Li et al., 2011; Brière et al., 2011; Burk et al., 2012; Eisenberg et al., 2014; Gupte, D’Costa & Chaudhuri, 2020; Cavazos-Rehg et al., 2021; Henneberger, Mushonga & Preston, 2021; Soriano-Sánchez & Jiménez-Vázquez, 2022).

Therefore, from the items and scales in Table 2, as control variables linked with adolescents’ environments, we define the variables displayed in Table 3.

The scales used to measure NORM, AGRESSIVE, SCHOOLENG, PARSUPP, and PARCONT were developed by Planet Youth (Planet Youth, 2018) and have been employed in numerous studies, both within reports on young people’s lifestyles by the organization and in peer-reviewed publications (for a comprehensive list, see https://planetyouth.org/the-method/publications).

The variables related to the information provided by the surveyed individuals, as presented in Table 2, were categorized into two groups. The first type of variable measures the perceived quality of information that the adolescent possesses regarding substance use (INF_LEVEL) and is derived directly from IQ9. The second group of variables measures the number of information sources (monitored and not monitored) from which such information originates (IQ10–IQ15). The results are presented as normalized values in Table 3.

Data analysis

To answer the RQ, we first measured the reliability of the NORM, AGRESSIVE, SCHOOLENG, PARSUPP, and PARCONT questionnaires by using conventional measures such as Cronbach’s alpha (α), convergent reliability (CR), and average variance extracted (AVE). The scales were considered reliable if α, CR > 0.7, and AVE > 0.5. Subsequently, we performed a regression analysis on the impact of the explanatory variables on the NUMSUBS.

Since we aim to explain a count variable, a possible alternative is to model it using a Poisson or negative binomial type model. The adjusted coefficients for each variable allowed us to measure the sensitivity of the average increase in the number of substances consumed in relation to the variations in the explanatory factors. The obtained results will be useful, but they do not capture the possibility that there may be a different sensitivity of the response variable to the change from nonconsumers to consumers of a single substance compared to the change from being a nonconsumer to being a polyuser.

To capture the possible difference in the sensitivity of the response as the number of substances consumed increases, we conducted a multinomial regression. We consider the baseline category to be nonconsumers. Consequently, the fitted coefficients for each variable provide information about the influence of each variable on the consumption of one, two, or three substances compared with nonconsumption. This approach allowed us to assess how the influence of each variable evolves as the number of substances consumed by adolescents increases.

The multinomial regression equation was adjusted hierarchically in two stages. First, we introduce the control variables. Subsequently, we fit the NUMSIBS by introducing variables related to the information available to the adolescents. Therefore, to assess whether incorporating the set of information-related variables is suitable, we analysed and compared the values of Akaike’s information criterion, Schwartz’s criterion, and the Hannan–Quinn criterion for both models. If the inclusion of these informative variables leads to a reduction in the values of the Akaike, Schwartz, and Hannan-Quinn measures, the result is considered adequate. To address the missing data, we used surveys whose items were completely fulfilled.

Results

Descriptive statistics

As shown in Table 2, 502 teenagers (38.41%) were admitted to having consumed alcohol, 196 (15%) had consumed tobacco, and 88 (7.28%) had consumed cannabis in the last 30 days. Likewise, Table 2 shows that while 51.10% of the sample stated that they had not consumed any of the studied substances in the last 30 days, 380 (29.07%) acknowledged consuming one substance, 117 (8.95%) admitted to consuming two, and 57 (7.80%) had consumed three.

In terms of information variables, Table 2 shows that nearly 80% of the respondents mostly agreed or completely agreed that they were well informed about the consequences of substance use. Concerning sources of information, the most relevant ones, in order of decreasing relevance, were schools (866, 66.26%), the Internet (861, 65.88%), and parents or legal guardians (815, 62.36%). The three least common sources, from least to most reported, were siblings (305, 23.41%), peers (584, 44.68%), and the mass media (720, 55%).

Similarly, within the pattern of information source usage, while the most common practice regarding monitored sources was obtaining information from all three monitored sources (451 responses), the least common practice was not recognizing any supervised source (121 responses). In terms of nonmonitored sources, the most common was recognizing one source (449 responses), and the least frequent was recognizing information from all three analysed sources (183 responses).

Figure 1 shows that among the adolescents who consumed only one substance, the majority consumed alcohol (26.01%), while the majority consumed tobacco (2.60%) or cannabis (0.46%). Within the group of adolescents who used two substances, the majority combined alcohol and tobacco (7.04%), those combining alcohol and cannabis (0.92%), and those combining tobacco and cannabis (0.99%) were notably lower. Notably, cannabis consumption was more common when cannabis was combined with tobacco and alcohol (4.36%) than when cannabis was used alone or in combination with either tobacco or alcohol.

Figure 1 Substance consumption patterns in our sample last 30 days.

Assessment of multinomial regression

Table 4 shows that, in general, all the scales exhibit reliability. This conclusion is especially evident for parental control, peer influence, and aggression, as indicated by Cronbach’s alpha (α) and composite reliability (CR) values both exceeding 0.7 and the average variance extracted (AVE) being above 0.5. For NORMS and SCHOOLENG, the AVE is slightly lower than 0.5. However, despite not being an optimal result, the scale’s consistency can still be accepted because the CR significantly exceeds 0.6 (Lam, 2012). Furthermore, the factor loadings of the indicators surpass 0.6 in all the cases, which supports the scale’s consistency.

Table 4 Scale reliability measurement of submission to norms, irritability, school support, parental control and peer influence.

	Cronbach α	CR	AVE	
Not adherence to norms	0.80	0.86	0.46	
Aggressiveness	0.83	0.88	0.59	
School engagement	0.74	0.83	0.49	
Parental support	0.864	0.905	0.66	
Parental control	0.91	0.93	0.61	
Peer influence	0.92	0.93	0.81	

Table 5 shows that the model that considers only the control variables is significant, as the LR ratio indicates that the inclusion of these variables significantly improves the explanatory capacity of the model compared to considering only a constant as an explanatory variable (LR ratio = 174.93 with p < 0.0001). However, the inclusion of the information variables improves the quality of the multinomial fit for all the information criteria used, as they always decrease in value. The Schwartz criterion ranged from 2,004.21 to 1,999.97, the Akaike criterion ranged from 1,873.63 to 1,826.37, and the Hannan–Quinn criterion ranged from 1,923.43 to 1,892.62. Similarly, the expanded model was also significant (LR-ratio = 217.61, p < 0.0001). Therefore, we limit ourselves to discussing the results of the fit that includes the information variables.

Table 5 Results of the multinomial regression estimates.

NUMSUBS = 1	
Model without information variables	Model including information variables	
Individual	OR	p value	95% CI	Individual	OR	p value	95% CI	
FEMALE	1.127	0.4514	[0.825–1.540]	FEMALE	1.114	0.5089	[0.809–1.533]	
AGE	2.268**	<0.0001	[1.672–3.076]	AGE	2.302**	<0.0001	[1.688–3.140]	
NORMS	1.156	0.0735	[0.986–1.355]	NORMS	1.122	0.1673	[0.953–1.320]	
AGRESSIVE	1.247**	0.0094	[1.056–1.473]	AGRESSIVE	1.252**	0.0095	[1.056–1.484]	
Microsystem	OR	p value	95% CI	Microsystem	OR	p value	95% CI	
SCHOOLENG	0.946	0.5129	[0.802–1.116]	SCHOOLENG	0.964	0.6733	[0.814–1.142]	
PARSUPP	0.950	0.5543	[0.801–1.127]	PARSUPP	0.955	0.6061	[0.802–1.137]	
PARCONT	1.310*	0.0441	[1.007–1.703]	PARCONT	1.328*	0.0419	[1.010–1.746]	
PEERINFL	1.323**	0.0053	[1.087–1.610]	PEERINFL	1.385**	0.0023	[1.124–1.707]	
					Information	OR	p value	95% CI	
					INF_LEVEL	1.390	0.3618	[0.685–2.823]	
					NON_SUP_S	2.259**	0.0021	[1.343–3.798]	
					SUP_S	0.499**	0.0097	[0.295–0.845]	
NUMSUBS = 2	
Model without information variables	Model including information variables	
Individual	OR	p value	95% CI	Individual	OR	p value	95% CI	
FEMALE	0.760	0.2817	[0.461–1.252]	FEMALE	0.690	0.1615	[0.411–1.160]	
AGE	3.780**	<0.0001	[2.301–6.210]	AGE	3.755**	<0.0001	[2.255–6.254]	
NORMS	1.811**	<0.0001	[1.382–2.373]	NORMS	1.827**	<0.0001	[1.382–2.416]	
AGRESSIVE	1.535**	0.0005	[1.206–1.953]	AGRESSIVE	1.524**	0.0008	[1.190–1.951]	
Microsystem	OR	p value	95% CI	Microsystem	OR	p value	95% CI	
SCHOOLENG	0.743*	0.024	[0.574–0.962]	SCHOOLENG	0.733*	0.0212	[0.563-0.955[	
PARSUPP	0.939	0.609	[0.740–1.193]	PARSUPP	0.969	0.8036	[0.757–1.241]	
PARCONT	0.848	0.3571	[0.598–1.204]	PARCONT	0.822	0.2951	[0.569–1.187]	
PEERINFL	1.825**	<0.0001	[1.454–2.289]	PEERINFL	1.938**	<0.0001	[1.519–2.471]	
					Information	OR	p value	95% CI	
					INF_LEVEL	1.833	0.2861	[0.602–5.584]	
					NON_SUP_S	4.907**	0.0002	[2.126–11.327]	
					SUP_S	0.233**	0.0004	[0.104–0.522]	
NUMSUBS = 3	
Model without information variables	Model including information variables	
Individual	OR	p value	95% CI	Individual	OR	p value	95% CI	
FEMALE	0.469	0.039	[0.229–0.962]	FEMALE	0.462*	0.0405	[0.221–0.967]	
AGE	4.635**	<0.0001	[2.300–9.343]	AGE	4.733**	<0.0001	[2.301–9.734]	
NORMS	1.662**	0.0079	[1.143–2.417]	NORMS	1.564**	0.0196	[1.074–2.278]	
AGRESSIVE	1.279	0.1504	[0.915–1.787]	AGRESSIVE	1.260	0.1877	[0.894–1.775]	
Microsystem	OR	p value	95% CI	Microsystem	OR	p value	95% CI	
SCHOOLENG	1.185	0.3184	[0.849–1.656]	SCHOOLENG	1.172	0.3581	[0.835–1.644]	
PARSUPP	0.890	0.4699	[0.648–1.221]	PARSUPP	0.904	0.5434	[0.654–1.251]	
PARCONT	1.010	0.9702	[0.604–1.687]	PARCONT	0.996	0.9881	[0.589–1.686]	
PEERINFL	2.151**	<0.0001	[1.666–2.777]	PEERINFL**	2.234**	<0.0001	[1.698–2.940]	
					Information	OR	p value	95% CI	
					INF_LEVEL	0.894	0.8729	[0.227–3.523]	
					NON_SUP_S	7.389**	0.0006	[2.376–22.978]	
					SUP_S	0.252**	0.013	[0.085–0.747]	
Schwartz	2,004.207				Schwartz	1,999.968				
Akaike	1,873.628				Akaike	1,826.369				
Hannan-Quinn	1,923.429				Hannan-Quinn	1,892.622				
LR-Ratio	174.93**				LR-Ratio	217.62**				
Note:

“*” and “**” denote statistical significance levels of, 5% and 1%, respectively.

Table 5 shows that age is a significant factor, and its influence increases as the number of substances consumed by teenagers increases. Thus, for NUMSUBS = 1, the odds ratio (OR) was 2.303, which increased to 3.755 for NUMSUBS = 2 and 4.733 for all the considered substances consumed by teenagers (all p < 0.0001). Aggressiveness is relevant for explaining the transition from not using substances to consuming one substance (OR = 1.252, p = 0.0095) and for the polyconsumption of two substances (OR = 1.525, p < 0.0008). However, AGRESSIVE is not relevant for explaining NUMSUBS = 3. On the other hand, rebelliousness is not relevant for explaining the prevalence of a single substance but is significant for explaining poly drugs, as for NUMSUBS = 2, OR = 1.827 (p < 0.0001) and NUMSUBS = 3, OR = 1.564 (p = 0.0196)). Finally, sex was relevant only for the consumption of the three substances. Being female protected against the combined consumption of alcohol, tobacco, and cannabis (OR = 0.462; p = 0.0405) but not against the consumption of two substances or one substance.

Regarding the contextual variables of young people’s microsystems, Table 4 shows that only the influence of peers has a clear impact on both single consumption and polyconsumption. As the number of substances consumed increased, the impact of PEERINFL increased. Thus, for NUMSUBS = 1, OR = 1.385 (p = 0.0023); for NUMSUBS = 2, OR = 1.938 (p < 0.0001); and finally, for NUMSUBS = 3, OR = 2.234 (p < 0.0001). Engagement at school may have some protective effect on the consumption of two substances (OR = 0.733, p = 0.0212), and parental control may facilitate the consumption of one substance (OR = 1.328, p = 0.0419). However, in general, variables related to parental style and support for teenagers’ perceptions of receiving from school were not relevant in explaining the patterns of drug consumption analysed.

Regarding the variables related to substance use, we observed that the degree of information provided by teenagers was not a significant factor in explaining the studied consumption patterns. In contrast, the number of formal and informal sources of information was consistently significant. Regarding unsupervised sources, as the number of substances consumed increases, their facilitating impact consistently increases. Thus, for NUMSUBS = 1, OR = 2.259 (p = 0.0021); for NUMSUBS = 2, OR = 4.907 (p = 0.0002); and for NUMSUBS = 3, OR = 7.389 (p = 0.0006). However, a higher number of supervised sources used in teenagers’ information results in protection against the prevalence of both one substance and more than one substance. This inhibitory effect decreased from the consumption of one substance (OR = 0.499, p = 0.0097) to the consumption of two substances (OR = 0.233, p = 0.0004). This protection also had a significant effect on the consumption of these three substances (OR = 0.252, p = 0.013).

Discussion

This study examined the primary patterns of polydrug use among adolescents and their key drivers. Although alcohol is the most commonly used single substance, tobacco tends to be used in conjunction with alcohol, and cannabis users often use both alcohol and cigarettes.

When assessing the explanatory factors of polydrug use, this study placed significant emphasis on the quality and quantity of information perceived by adolescents, as well as the number of information sources. It distinguishes between controlled and uncontrolled sources. The influence of these factors is controlled for variables related to the individual characteristics of the adolescent, such as age, sex, or temperament, as well as the primary variables associated with their microsystem (school, family, and peers).

Although the examined individual and environmental variables are well established in the literature, investigations into the impact of variables related to the level of information and available sources among adolescents are groundbreaking. Our findings indicate that the inclusion of additional variables enhances the explanatory power of multiple drug consumption patterns.

We should differentiate the findings related to the control variables (individual and environmental) from those that are subject to special analysis, which are related to the information available to teenagers and their sources. The utilization of multinomial regression instead of Poisson-type count regression has enabled us to quantify not only the significance and direction of the relationship between the number of substances consumed and the explanatory variables but also the varying intensity of their impact and significance based on the number of substances consumed by the adolescent.

The most relevant factor for the individual control variables was the age of the teenagers. We observed that age was positively related to all the substances consumed (one, two, or three substances), and its influence increased as the number of substances consumed increased. The results, in any case, are logical, as we observed that the number of teenagers initiating the use of all substances increased with age. Thus, the probability of a teenager being a polydrug user increases with age. Therefore, our findings align with those of Libuy, Ibáñez & Mundt (2020) and Andres-Sanchez & Belzunegui-Eraso (2022), who demonstrated that the probability of consuming cannabis increases with age, and with those of Brière et al. (2011) and Jongenelis et al. (2019), who observed a greater prevalence of the use of more than one drug as youth age.

The fact that being male significantly facilitates the consumption of three substances, but not fewer, is consistent with the variability of the sex variable in studies of the prevalence of substance use by adolescents (McHugh et al., 2018). On the one hand, there are physiological differences between men and women in their reactions to any substance (National Institute on Drug Abuse, 2022), and men typically tend to be heavier substance users (McHugh et al., 2018). However, the latter assertion depends on the culture of the society being analysed; the more similar the gender roles are, the fewer differences in substance use tend to be (McHugh et al., 2018). In some cases, women may even consume more tobacco than males, as shown in studies by Eisenberg et al. (2014) and Andres-Sanchez, Belzunegui-Eraso & Fernández-Aliseda (2021). On the other hand, men may have greater engagement with cannabis (Francis et al., 2019; Andres-Sanchez & Belzunegui-Eraso, 2022), tend more toward polydrug use (Brière et al., 2011) and are more likely to develop disorders associated with such practices (McHugh et al., 2018).

We found that while the aggressiveness of teenagers is relevant to their substance use, this finding is in line with the reports of various authors that higher levels of irritability are an enabler of substance use (Picciotto et al., 2015; Kivimäki et al., 2014; Turner et al., 2018; Francis et al., 2019; Nawi et al., 2021). Similarly, rebelliousness has a significant positive impact on strict polydrug use (two or three substances), which is supported by the findings of Boddington & McDermott (2013), Kwon et al. (2018), and Nawi et al. (2021) but not on the use of a single drug.

Among the control variables in the microsystem of the teenagers’ environment, the most relevant factor in the consumption of multiple substances was peer influence. In fact, this factor progressively increases in impact, as teenagers tend to consume more substances. These results are consistent with the literature reviewed in this study (Cengelli et al., 2012; Brière et al., 2011; Burk et al., 2012; Eisenberg et al., 2014; Gupte, D’Costa & Chaudhuri, 2020; Cavazos-Rehg et al., 2021; Henneberger, Mushonga & Preston, 2021; Soriano-Sánchez & Jiménez-Vázquez, 2022).

The main contribution of this study is the assessment of the effects of information variables on the consequences of substance use on behaviors related to the number of substances used in the last 30 days. Our findings reveal that an overwhelming majority of adolescents (approximately 80%) tend to either agree or fully agree with the statement “Do you consider yourself adequately informed about the repercussions of substance consumption?” However, it is crucial to note that the lack of statistical significance concerning the level of information on substance use implies that this information does not serve as a distinguishing element capable of elucidating the varying degrees of substance use among adolescents. These findings contrast with those of Dermota et al. (2013) and Belzunegui-Eraso et al. (2020), who indicated that greater amounts of information lead to a greater tendency toward drug use, as well as with the findings of Fleary, Joseph & Pappagianopoulos (2018) and Sadeghi et al. (2019), which indicated that greater health literacy inhibits smoking habits. However, we understand that this does not contradict the analysis conducted, as it reconciles both positions.

However, obtaining information from sources that are presumably more reliable, such as schools, parents, and mass media, which highlight the risks associated with drug consumption, can positively impact health literacy and potentially safeguard adolescents against polydrug use. These channels often emphasize the dangers of substance use, which is a significant factor in protecting against drug use (Zimmerman & Farrell, 2017). This protective effect may increase with the number of substances consumed. In contrast, information acquired from sources that are less reliable and controllable, such as peers, the Internet, and siblings (Chen et al., 2018), has the potential to negatively affect health literacy and facilitate polydrug use. This enabling effect may be amplified by the number of substances used.

The findings of this study have valuable implications for health authorities and organizations responsible for designing preventive interventions for substance use among young people. While greater information about substance use may not necessarily lead to a lower prevalence of polydrug use habits, it is crucial for health authorities to monitor sources of information about substance use and ensure that reliable information is provided. Authorities should also establish interventions that enable teenagers to access useful resources for their health while protecting them from harmful and unreliable information, particularly on platforms such as the Internet. These interventions should also protect them from influences, such as peers or siblings, whose testimonies may emphasize issues related to pleasure or prestige rather than the more relevant ones, which are the harmful effects outlined in the introduction of the study.

However, caution must be exercised when generalizing our results to other places because our study was conducted in a specific geographical location with its own unique sociodemographic characteristics. Tarragona, a city with approximately 150,000 residents, has a distinct economic landscape characterized by industry and services. It is also home to a significant number of migrants from South America and North Africa. Therefore, while our findings provide valuable insights applicable to European regions with similar social and regulatory settings and population demographics, they may not directly apply to areas with different characteristics. For example, regions with fewer migrant populations or economies, primarily based on the primary sector, may exhibit different dynamics regarding adolescent smoking habits and sources of information.

It should also be understood as a limitation of the study that the outcome reflects the number of substances consumed by adolescents in the last 30 days, but it does not capture the intensity with which they were consumed. Subsequent analyses could be differentiated not only by whether a substance is consumed but also by the degree to which it is consumed in combination with others, which can also be used sporadically or habitually.

Furthermore, as we consider the future implications of our study, we must remain vigilant about the ever-evolving landscape of information dissemination. The digital age has brought new challenges and opportunities, especially as the Internet has become a major information source. As regulations regarding online information evolve and the behaviors of adolescents continue to adapt, our study serves as a crucial snapshot in time. Therefore, the insights gained from our research are invaluable but should be interpreted with caution.

Conclusions

In this study, we conducted a cross-sectional survey in Tarragona, Spain, to investigate the relationships between information sources and the consequences of substance and polydrug use among adolescents. Our research revealed a notable pattern: An increased number of monitored information sources had a substantial impact on discouraging substance use among adolescents. Conversely, a higher prevalence of unsupervised sources appears to facilitate the development of such habits. This phenomenon led us to speculate on the potential implications for public health and educational strategies. Monitored sources of information, which are subject to regulation by laws and public authorities, have emerged as trustworthy sources of knowledge about the harmful consequences of drug use. However, unmonitored sources operating beyond regulatory oversight represent a worrisome contrast. These sources have the potential to undermine health literacy, as they may disseminate inaccurate or misleading information, making it difficult for adolescents to discern the true risks associated with drug use.

Supplemental Information

Supplemental Information 1 Dataset with the survey that based the study.

Click here for additional data file.

Supplemental Information 2 Codes of items used in the paper.

Click here for additional data file.

Supplemental Information 3 STROBE-information.

Click here for additional data file.

Supplemental Information 4 Questions in English.

Click here for additional data file.

Supplemental Information 5 Questionnaire in Spanish.

Click here for additional data file.

Additional Information and Declarations

Competing Interests

Author Contributions

Human Ethics

Data Availability

The authors declare that they have no competing interests

Jorge de Andrés-Sánchez analyzed the data, prepared figures and/or tables, authored or reviewed drafts of the article, and approved the final draft.

Angel Belzunegui-Eraso conceived and designed the experiments, performed the experiments, prepared figures and/or tables, authored or reviewed drafts of the article, and approved the final draft.

Francesc Valls-Fonayet conceived and designed the experiments, performed the experiments, authored or reviewed drafts of the article, funding, and approved the final draft.

The following information was supplied relating to ethical approvals (i.e., approving body and any reference numbers):

The University Rovira i Virgili granted Ethical approval to carry out the study within its facilities CEIPSA-2021-PRD-0039.

The following information was supplied regarding data availability:

The data is available in the Supplemental File.

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
