# Peer review of "The significance of information variables in polydrug use by adolescents: insights from a cross-sectional study in Tarragona (Spain)"

_PeerJ, doi:10.7717/peerj.16801_

## Round 0.1 · original submission · Major Revisions

The reviewers underlined that the study is interesting. Nevertheless, they described the need for a major revision to support the strength of the paper and demonstrate its originality.

The comment about regression models also needs to be carefully considered.

Reviewer 1 ·

Basic reporting

.

Experimental design

.

Validity of the findings

.

Additional comments

1. …”while also increasing the likelihood of future use of harder drugs due to the Gateway effect”
-The authors should really explain what they mean by this. Evidence that these drugs actually cause individuals to use hard drugs in the manner the gateway hypothesis posits is pretty scant. Instead, this is often via social factors like potentially facilitating access and the stigma associated with hard drug use being reduced among those who use these soft drugs.

2. “It is well known that the perception of the potential harms of drug use deters adolescents from using them (Zimmerman & Farrell, 2017).”
-I would recommend tempering this statement if it cannot be supported by more than one citation. A lot of research in the US suggests that deterrence-based policy and programming has done little to reduce drug use prevalence. The DARE program and the War on Drugs policies are prime examples of this.

3. “Peers and siblings often embrace a perspective that prioritizes immediate gratification and hedonism over long-term risks and may center around themes such as enhancing social enjoyment, relaxation, or associating substance use with status and glamour.”
-Might also be worth discussing how cognitive development (the dual systems model, specifically) helps to explain why adolescent decision-making tends to prioritize immediate gratification over long-term risk of substance use.

4. Polysubstance use needs to be more clearly defined in terms of how it will be operationalized in the study within the literature review. Simply stating that use of two substances in a given period of time is unclear in this regard. Is this concurrent use or is it just use of two drugs within an observation period?

5. Why would the authors use multinomial logistic regression over a negative binomial or poisson regression model here? The dependent variable appears to be a count of the number of substances that participants reported using and has a linear scale with discrete units of measurement between each category.

·

Basic reporting

no comment

Experimental design

no comment

Validity of the findings

no comment

Additional comments

This is a good study addressing the substance use pattern in adolescents and its determinants. The questionnaire/scale is well-designed and reliable. In addition to personal and environmental factors, the authors uniquely incorporate information source variables to construct the explanatory model. The underlying rationale is well elucidated.

Consistent with previous literature and our knowledge to date, the findings are well interpreted and discussed. I think the authors really did a good job and brought valuable implications for prevention of substance use.

The following are my small suggestions:

1. Conclusion should be concise. The writing about strengths and limitations of this study should be placed in the Discussion section.

2. The authors should read the manuscript again carefully and some minor mistakes should be corrected. For examples, Line 49 should begin with "and in other countries" and Line 216 should end with a period.

---

## Round 0.2 · accepted · Accept

I am pleased to inform you that your manuscript is accepted for publication in PeerJ.